# Out of East Asia: Early Warning of the Possible Invasion of the Important Bean Pest Stalk-Eyed Seed Bug *Chauliops fallax* (Heteroptera: Malcidae: Chauliopinae)

**DOI:** 10.3390/insects14050433

**Published:** 2023-05-01

**Authors:** Yanfei Li, Juhong Chen, Shujing Wang, Kun Jiang, Jiayue Zhou, Runqi Zhu, Cuiqing Gao, Wenjun Bu, Huaijun Xue

**Affiliations:** 1Institute of Entomology, College of Life Sciences, Nankai University, Tianjin 300071, Chinawangsj@nankai.edu.cn (S.W.);; 2Co-Innovation Center for the Sustainable Forestry in Southern China, College of Forestry, Nanjing Forestry University, Nanjing 210037, China

**Keywords:** *Chauliops fallax*, population genetics, ecological niche modelling, invasive biology

## Abstract

**Simple Summary:**

The stalk-eyed seed bug *Chauliops fallax* is an important agricultural pest of soybean and was first reported to occur outside East Asia. Here, the native evolutionary history, recent invasion history, and potential invasion threats of *C. fallax* were reported for the first time based on population genetic methods and ecological niche modelling. Four native genetic groups (EA, WE, TL, and XZ) and an east–west differentiation pattern consistent with the geographical characteristics of three-step landforms were well supported in China. The Kashmir sample was found to come from a recent invasion event, and the invasive source was determined to be the populations of the coastal areas of southern China. North America might have high a risk of invasion, which poses a serious threat to local soybean production. Our results provided new insights into the monitoring and management of *Chauliops fallax* in the early invasion stage.

**Abstract:**

The short stay at the beginning of the invasion process is a critical time for invasive species identification and preventing invasive species from developing a wider distribution and significant economic impact. The stalk-eyed seed bug *Chauliops fallax* is an important agricultural pest of soybean and was first reported to occur outside East Asia. Here, we reported the native evolutionary history, recent invasion history, and potential invasion threats of *C. fallax* for the first time based on population genetic methods and ecological niche modelling. The results showed that four native East Asian genetic groups (EA, WE, TL, and XZ) were well supported, showing an east-west differentiation pattern consistent with the geographical characteristics of three-step landforms in China. Two main haplotypes existed: Hap1 might have experienced a rapid northwards expansion process after the LGM period, and Hap5 reflected local adaptation to the environment in southeastern China. The Kashmir sample was found to come from the recent invasion of populations in the coastal areas of southern China. Ecological niche modelling results suggested that North America has a high risk of invasion, which might pose a serious threat to local soybean production. In addition, with future global warming, the suitable habitat in Asia will move towards the higher latitude region and gradually deviate from the soybean planting area, which indicates the threat of *C. fallax* to soybean production in Asia will decrease in the future. The results could provide new insights into the monitoring and management of this agricultural pest in the early invasion stage.

## 1. Introduction

With the intensification of the global integration process, trade exchanges between different regions are increasing, and there are increasing opportunities for species to be transported beyond their native range, therefore becoming invasive species [1]. Biological invasion is one of the main causes of global biodiversity loss and global ecosystem function erosion [2,3]. At the same time, it also leads to negative impacts on human life and health [4,5], agriculture [6], and food security [7]. Studies on the history of many invasions have shown that species experience a short stay in their initial invasion sites before forming a large invasion threat, eventually producing a bridgehead effect [8,9,10,11] or an overt bottleneck effect [12,13,14,15,16]. These effects will then trigger a self-accelerating process of invasion [1]. Additionally, it is controversial whether it is due to mutations that have gained a strong invasive ability or due to multiple admixture events [1,17]. However, it is worth noting that whichever reasons enhance the invasion ability, the short stay at the beginning of the invasion process exists for many invasive species [18,19]. Therefore, we believe that this period will be a critical time for invasive species identification to prevent invasive species from developing a wider distribution and significant economic impact.

If we want to be more precise about invasive species, we need to start with two important questions: where do they come from and where will they go [20]? For the first question, we need to clarify the invasion history of the species that have been invaded and clarify the source and process of invasion. This will help us eliminate it from the source and avoid invasion leakage at each early invasion site. The population genetics method has been widely applied to the study of species and population-level evolutionary history [14,21,22,23]. By combining population genetics with invasion biology, the source population of invasive species can be accurately identified, and the invasion history can be clearly explained, providing a solid evolutionary biological basis for the early identification of invasive species and avoidance of further invasion [23,24,25]. For the second question, we need to understand where invasive species are more likely to invade to strengthen quarantine and prevention and control to prevent the entry of invasive species. The ecological niche model (ENM) is based on the conservative nature of a species’ ecological niche, and information on the current climatic environment of that species’ distribution is used to predict what other possible invaded areas exist globally under the current climatic conditions of the species and the future distribution trends of the species under future climate change conditions [26,27,28,29]. To date, ENM has played a good role in the early warning of the invasion of many invasive species [30,31].

The stalk-eyed seed bug *Chauliops fallax* (Scott, 1874) is an important legume pest that mainly feeds on legumes such as kudzu and soybeans [32,33,34,35]. Studies have shown that it can act throughout the growth period of the crop, causing a 10–30% yield reduction in legumes [32,33]. Scott established *C. fallax* in 1874 based on specimens from the Japanese region. According to the literature, it is mainly naturally distributed in East Asia (mainly in China, Korea, and Japan) (Catalogue of the Palaearctic Heteroptera: https://catpalhet.linnaeus.naturalis.nl/linnaeus_ng/app/views/introduction/topic.php?id=9&epi=1, accessed on 20 January 2023) [35]. Through field collection, we also found *C. fallax* in Thailand. *C. fallax* has also been recorded in the Indian subcontinent by Chopra & Rustagi, 1982, but it was corrected by Sweet & Schaefer, 1985, according to the obvious morphological difference between Indian and East Asian specimens, and was described as a separate species: *C. choprai*. Therefore, no credible distribution record was available for *C. fallax* outside East Asia until Ashika et al. uploaded a cytochrome *c* oxidase subunit I (*COI*) molecular fragment of *C. fallax* from Kashmir on NCBI (GenBank: MN584895) in 2020 [35,36,37]. According to the preliminary judgment of the sequence data, the record is the real *C. fallax.* This aroused our vigilance to determine whether the stalk-eyed seed bug was originally found in this area or whether it was the result of a recent invasion.

In this study, we first reconstructed its population evolution history in its native range through the population genetics method based on *COI* (in order to match the Kashmir sample, we chose *COI* as the molecular marker). Then, we analyzed the source of the Kashmir sample (local distribution or recent invasion) and the possible invasion history. Finally, we used ENM to predict the current and future potential distribution areas of *C. fallax* and explored its potential threat to soybean production by overlapping with soybean-planted areas.

## 2. Materials and Methods

### 2.1. Molecular Data Acquisition and Analysis

#### 2.1.1. Sampling and Laboratory Procedures

A total of 113 samples from 18 populations of *C. fallax* from China and Thailand were used for molecular experiments (Figure 1A and Table 1). Samples were preserved in 100% ethanol and stored at −20 °C until used. DNA was extracted from muscle tissue using the Universal Genomic DNA Kit (CWBIO, Beijing, China). Polymerase chain reaction (PCR) was used to amplify the *COI* gene barcoding region. The primers (Cfal1315F: 5′-AACATTGACTATAAAGCCTTGATAAGAGGT-3′; Cfal3044R: 5′-AATATTGATAGTTGTTCTATTACTGGCGAT-3′) were designed based on the mitochondrial genome of *C. fallax* (GenBank: NC_020772). The PCR procedure included an initial denaturation at 94 °C for 2 min, followed by 31–33 cycles of 30 s at 92 °C, 30 s at 53 °C and 1.5 min at 72 °C, ending with a final extension at 72 °C for 8 min. PCR products were checked with 1% agarose gel electrophoresis and sent to the Beijing Genomics Institute (BGI) for Sanger sequencing in both directions using an ABI3730XL Sequencer. One *COI* dataset from Kashmir was downloaded from NCBI (GenBank: MN584895). Sequences were visually proofread in BioEdit v7.2.5 and aligned in Mafft v7.037 [38].

#### 2.1.2. Species Identification, Genetic Polymorphism, Phylogenetic Analysis and Population Genetic Structure

The sample of Kashmir only has molecular data, without any morphological pictures or descriptions; therefore, we need to use the barcoding method to identify the species first. The genetic distance estimations based on *COI* were calculated in MEGA X [39] using the Kimura 2-Parameter (K2P) substitution model, 1000 bootstrap replicates, and the “pairwise deletion” option for missing data.

Genetic diversity was estimated for each location and the entire sample as the number of haplotypes (Hn), haplotype diversity (Hd), and nucleotide diversity (π), which were calculated in DNASP v5.1 [40]. Phylogenetic analysis was performed based on *COI* using the neighbour-net method implemented in SplitsTree v4.14.5 [41]. A haplotype network was constructed in PopART v1.7 [42] with the option “Median Joining network”.

The 18 native populations were used for the analysis of the population genetic structure. Two approaches were applied to gain further insight into the population genetic structure. First, a Bayesian analysis of population structure was represented in BAPS v6.0 [43] with the option “Spatial clustering of groups”, which combined sample locations with the likelihood of the genetic data and estimated the best K values (maximum number of genetically diverged groups). Second, the Spatial Analysis of Molecular Variance was implemented in SAMOVA v2.0 [44]. This method is based on a simulated annealing procedure that aims at maximizing the proportion of total genetic variance due to differences between groups of populations. SAMOVA v2.0 [44] was run using 100 simulated annealing processes for K = 2–10 clusters. The genetic differentiation between the populations and the groups (defined by phylogenetic analysis) was calculated based on the pairwise difference method in ARLEQUIN v3.5.2.2 [45].

#### 2.1.3. Historical Population Dynamics

The historical population dynamics were estimated in BEAST v2.5 [46] using a Bayesian coalescent-based method (Bayesian Skyline Plot, BSP) based on the native samples. This method was used to reconstruct the effective population size fluctuations since the time of the most recent common ancestor (TMRCA). The MCMC chains were run for 1000 million generations until the ESS value was larger than 200. A relaxed uncorrelated lognormal molecular clock with a mutation rate of 0.6–1%/per million years [47] was used. Demographic history was reconstructed in Tracer v1.7 [48]. In addition, two neutrality tests, i.e., Tajima’s D and Fu and Li’s D were calculated in DNASP v5.1 [40].

### 2.2. Distribution Data Acquisition and Analysis

#### 2.2.1. Species Occurrence Data and Climate Data

*Chauliops fallax* occurrence data were collected from (1) collection records: the College of Life Sciences, Nankai University (NKU, Tianjin, China) has more than 70 years’ worth of specimens in its collection; (2) the Global Biodiversity Information Facility (GBIF, https://www.gbif.org/, accessed on 20 January 2023); (3) Integrated Digitized Biocollections (iDigBio, https://www.idigbio.org/, accessed on 20 January 2023) and (4) iNaturalist (iNaturalist, https://www.inaturalist.org/, accessed on 20 January 2023). The occurrence records were cleaned by removing duplicate observations and observations without latitudes and longitudes. The geographic coordinates were filtered out by enforcing a distance of 30 km between records by the *Wallace* package in R [49], as ecological niche models are sensitive to sample bias [50]. Finally, 83 unique records of *C. fallax* were obtained (Appendix A).

A total of 19 bioclimatic variables with a spatial resolution of 2.5 min were obtained from the WorldClim (https://worldclim.org/, accessed on 20 January 2023) website for both historical (1970–2000) and future conditions [51]. The cross-correlations among the bioclimatic variables and the results of the analysis of variable contributions were combined to select bioclimatic variables for *C. fallax*. Multicollinearity among the bioclimatic variables was reduced by selecting only one variable that ranked first in the analysis of variable contributions from a set of highly correlated variables (|r| ≥ 0.8). Finally, 10 bioclimatic variables were selected for further analysis: mean diurnal range (BIO2), isothermality (BIO3), temperature seasonality (BIO4), max temperature of warmest month (BIO5), minimum temperature of coldest month (BIO6), mean temperature of wettest quarter (BIO8), annual precipitation (BIO12), precipitation seasonality (BIO15), precipitation of warmest quarter (BIO18), and precipitation of coldest quarter (BIO19). We used datasets of bioclimatic variables for current (1970–2000), 2050 (2041–2060) and 2070 (2081–2100) scenarios to predict the current and future spread of *C. fallax*. Future distribution climate projections were evaluated from three earth system models (ESMs) that were part of the Coupled Model Intercomparison Project Phase 6 (CMIP6): (a) the Canadian Earth System Model v5 (CanESM5); (b) the Institute Pierre-Simon Laplace coupled model v6 (IPSL-CM6A-LR) and (c) the Max Planck Institute for Meteorology Earth System Model (MPI-ESM1-2-LR), using two different emissions scenarios (shared socioeconomic pathway: SSP): SSP126 and SSP585 in 2041–2060 (2060), 2061–2080 (2080), and 2081–2100 (2100), respectively. SSP126 and SSP585 have the lowest greenhouse gas emissions [52] and highest greenhouse gas emissions [53].

#### 2.2.2. Ecological Niche Modelling

We used maximum entropy niche modelling (MaxEnt v3.4.3) [54] to model the potential distribution of *C. fallax* under present and future climate change scenarios. Before modelling, the model-tuning procedure was performed using the R packages “ENMeval” [55]. The model with the lowest delta Akaike information criterion (delta AIC) value was selected as the best model for the modelling. In this study, output format = logistic, random test percentage = 25, regularization multiplier = 2, convergence threshold = 0.0001, and maximum number of background points = 10,000, which all were default parameter settings in MaxEnt except maximum iterations (default: 500). Seventy-five percent of the data were used for training, and 25% were used to test the model. We ran 15 replicates (regularization multiplier = 2; maximum number of background points: 10,000) involving the 10 environmental layers selected above.

We mapped the global potential distribution of *C. fallax* in Arcgis v10.8 software to produce a suitability map under present and future (i.e., 2060 and 2100) climate change conditions with two different emissions scenarios (SSP126 and SSP585). First, we removed the areas where the standard deviation was higher than the largest standard deviation of *C. fallax* occurrence to ensure conservative projected results. Then, the probabilities of habitat suitability for *C. fallax* were divided into the following four categories: 0.0-0.177681 as unsuitable habitat, 0.177681–0.34739 as poorly suitable habitat, 0.34739–0.404495 as moderately suitable habitat, and 0.404495–1 as highly suitable habitat. Categories were generated by extracting logistic values where species occurred, and the logistic value of the last 5% of locations was set as the threshold between unsuitable and poor suitability, which was 0.177681. In the same way, we obtained the thresholds between poor and moderate suitability and moderate and high suitability by sorting the last 10% (0.34739) and 15% (0.404495). The coverage area (in km^2^) of the potential distribution of *C. fallax* under the present climatic conditions was computed for each continent (Africa, Asia, Europe, North America, Oceania, and South America), taking into account the categories.

## 3. Results

### 3.1. The Phylogeographical Pattern and Demographic History of C. fallax in East Asia

The dataset was built using 633 bp *COI* and containing 114 samples from 19 populations. The 41 polymorphic sites included 16 singleton variable sites and 25 parsimony informative sites. A total of 30 unique haplotypes were derived. High haplotype diversity (>0.5) was observed in nine populations, and the JSNJ population had the highest haplotype diversity (0.933). The nucleotide diversities ranged from 0.00000 to 0.00274 (Table 1).

Four independent genetic groups (Figure 1) were identified by phylogenetic analysis, and these were named according to the area where the population was located: the east group (EA), west group (WE), Thailand group (TL), and Xizang group (XZ). The EA group was mainly located in the third step region of China, the WE group was located on the southern edge of the Sichuan Basin of the second step region of China, the XZ group was located in the first step region in China, and the TL group was located in the southeastern peninsula. The haplotype network showed that there was no shared haplotype among the groups (Figure 2A). There were two main haplotypes in the EA group: Hap1 and Hap5. Hap1 was mainly located in the North China Plain (Figure 2B) and appeared in seven populations (including AHFX, AHQM, JSNJ, SXQL, TJJX, ZJNB, and ZJQZ) (Figure 2B). Furthermore, Hap1 has formed a typical star-shaped haplotype network with the main haplotype as the core and with many derived haplotypes (Figure 2A). Hap5 was mainly located in the hilly areas of Southeast China and appeared in six populations (including AHQM, HBWH, HNYL, JSNJ, JXJL, and JXYF) (Figure 2C). There was one main haplotype in the WE group that appeared in four populations, including GZSY, GZZY, YNCX, and YNQJ (Figure 2D). FST values among the 19 populations ranged from −0.86339 to 1.00000 (Appendix A and Appendix A). FST values among the four groups ranged from 0.68597 to 0.81682 (Appendix A and Appendix A). In addition, the Kashmir sample had the lowest genetic difference from the Zhejiang samples (ZJNB & ZJQZ), and the genetic distance was only 0.37%. Additionally, the sample from Kashmir was clustered into the EA group and had a closer genetic relationship with Hap1 (with 2 mutated steps) (Figure 2A).

BAPS identified six clusters across 18 native populations (Figure 1A and Appendix A). Populations in the EA group were clustered into two clusters located in the south and north. The populations in the WE group were also grouped into two clusters, with SCLD alone becoming one. The TL and XZ groups emerged as separate clusters. According to the SAMOVAs (Appendix A and Appendix A), FCT continued to rise from K = 2 clusters to K = 10 clusters; FSC continued to decrease from K = 2 clusters to K = 10 clusters. Therefore, a reliable K value was not obtained by the SAMOVAs.

The values of Tajima’s *D* and Fu and Li’s *D** are not significant (Tajima’s *D* = −0.95545, *p* > 0.10; Fu and Li’s *D** = −2.13766, 0.10 > *p* > 0.05), which supported the neutral evolution of the populations. BSP reconstructions showed that the population size of *C. fallax* remained stable during the last glacial period, rapid demographic growth began approximately 5000 years ago, and a slight downward trend followed (Figure 3A). The population size of the EA group remained stable during the last glacial period, and the demographic slight growth occurred between approximately 10,000 years ago and 2500 years ago and has remained stable since then (Appendix A). The population size of the WE group remained stable during the LGM period, and slight demographic growth began approximately 2500 years ago (Appendix A).

### 3.2. Ecological Niche Modelling

#### 3.2.1. Model Performance and Influence of Bioclimatic Variables

A higher mean training AUC value of 0.8274 and a test AUC value of 0.8128 indicated a high accuracy of the MaxEnt model to predict the potential distribution of *C. fallax* (Appendix A). The most important bioclimatic variables for *C. fallax* prediction were BIO4 (39.9%), followed by BIO6 (18.5%), BIO5 (11.3%), and BIO8 (10.6%) (Figure 4 and Appendix A). These four variables were the strongest predictors of *C. fallax* distribution and contributed 80.3%. The model also showed that BIO18 (contributing 7.7%) is also important (Figure 4). When omitted, BIO18 was the environmental variable that decreased the gain the most and therefore appeared to have most of the information that was not present in the other variables. (Appendix A). Overall, the temperature was the main factor limiting the distribution of *C. fallax*.

#### 3.2.2. Current Potential Distribution of *C. fallax*

The map with habitat suitability scores for the occurrence of *C. fallax* at the global scale is shown in Figure 5. The MaxEnt model predictions showed that the global suitability of *C. fallax* is concentrated in Africa, Asia, Europe, North America, Oceania and South America, covering an area of approximately 11.30% of the globally available land. According to the modelling, the highly suitable habitat for *C. fallax* was found to be 3.26 × 10^6^ km^2^ (Appendix A). Asia and North America have the most favorable climatic conditions for *C. fallax* (Figure 5 and Appendix A). The greatest optimum suitable habitats were recorded in Asia and North America and were approximately 2.53 × 10^6^ km^2^ and 6.59 × 10^5^ km^2^, respectively. Moderately and poorly suitable habitat for *C. fallax* was found to be 1.91 × 10^6^ km^2^ and 1.17 × 10^7^ km^2^, respectively.

The overlapped map result (Appendix A and Table 2) indicated that the overlapped area was approximately 9.14 × 10^7^ km^2^ and was mainly located in Asia and North America (Figure 4). Asia, as the origin of *C. fallax*, has the highest overlapping area (3.71 × 10^6^ km^2^, Table 2) and the greatest optimum suitable habitat area overlapped with soybean planted area (2.04 × 10^6^ km^2^). Although Europe has an approximately 1.23 × 10^6^ km^2^ overlapping area between suitable *C. fallax* habitat areas and soybean planted areas, the moderate (2.32 × 10^4^ km^2^) and high (7.53 × 10^3^ km^2^) suitable habitats of *C. fallax* were low compared with those of North America (7.58 × 10^5^ km^2^, 4.87 × 10^5^ km^2^). Oceania has the lowest overlapping area (1.93 × 10^5^ km^2^).

#### 3.2.3. Predicted Future Potential Distribution of *C. fallax*

The results of the MaxEnt model under future climate change scenarios SSP126 and SSP585 for 2060 and 2100 are presented in Figure 6 (CanESM5), Appendix A (IPSL-CM6A-LR) and Appendix A (MPI-ESM1-2-LR). Suitable habitat for *C. fallax* was found to be moved poleward in the future for both the SSP126 and SSP585 emission scenarios. The suitable habitat area would increase by 2.67 × 10^5^ km^2^ (average results of the three models) and 4.88 × 10^5^ km^2^ for the SSP126 and SSP585 emission scenarios, respectively, in 2060 compared to the present suitable habitat area (Appendix A). However, the suitable habitat area for *C. fallax* would decrease by 1.57 × 10^5^ km^2^ and 9.85 × 10^6^ km^2^ for the SSP126 and SSP585 emission scenarios in 2100, respectively, relative to the present climate conditions.

## 4. Discussion

### 4.1. Population Structure and Population Dynamics History in the Native Range

East Asia has complex terrain with mountainous terrain structure characteristics, and the mountain environment provides an important geographical barrier for species differentiation in this area [56,57]. The consistency between the genetic group and the geographical features indicated the significant phylogeographical structure of *C. fallax* in East Asia. The obvious east-west differentiation pattern formed along the three-step landforms in China [57] indicated that mountains were an important geographical barrier promoting the genetic differentiation of *C. fallax*. The two main haplotypes (Hap1 and Hap5) presented a north-south differentiation pattern with the Yangtze River and the middle and lower reaches of the Yangtze River plain as the boundary. The population in this area showed typical characteristics of high Hd and low π (Table 1), indicating that the population of Hap1 has experienced a rapid diffusion process in North China. Hap5 was different from Hap1, with characteristics of low Hd and low π of populations, and the derived haplotypes were rare, which showed that Hap5 had strong adaptability to the environment of Southeast China and was dominant here.

The terrain environment in Southeast China is dominated by low mountains and hills, which provided important refuge for many creatures during the ice age [57,58]. Additionally, the ENM showed that the highly suitable region was mainly located in the hilly area of Southeast China during the LGM period. The historical dynamics of the population also showed that the population size of the species was stable during the LGM period. Therefore, we speculated that the ancestral population of the EA group was mainly distributed in Southeast China during the LGM period, and the mountain environment protected it from the climatic shock of the ice age [57,58,59]. 

Genetic diversity analysis showed that the HBWH population located at the junction of North China and Southeast China had the highest nucleotide diversity. Additionally, the nucleotide diversity of the population north of the Yangtze River decreased gradually from south to north. ENM also showed the continued northward expansion of the suitable area after the ice age. Therefore, we speculated that the Hap1 haplotype derived from the ancestral population (mainly located near the HWBH population) had strong adaptability and diffusion ability. After the LGM period, the Hap1 haplotype rapidly extended to North China, and many haplotypes were derived to adapt to environmental changes during the diffusion process. Hap5, which was also derived from the ancestral population, was more suited to the local hilly environment, so it formed a dominant haplotype in southeastern China. However, our speculation was only based on limited geographic populations (note especially the lack of samples from Korea and Japan) and *COI* fragments, which should be supported by a larger sample size and more molecular evidence in the future.

### 4.2. Invasion History Reconstruction of the Kashmir Sample

According to the genetic distance analyses, the Kashmir sample had the lowest genetic difference from the Zhejiang samples (ZJNB & ZJQZ), and the genetic distance was only 0.37%, indicating that the Kashmir sample was *C. fallax*. The Kashmir sample was clustered into the EA group and not with the geographically closer Thailand or Tibet and had no significant genetic differentiation with East Asian samples, and there were only two mutation sites different from Hap1. According to previous literature records, Sweet & Schaefer (1985) described the Indian “*C. fallax*” as a new species “*C. choprai*” and believed that there was no *C. fallax* outside East Asia [35,36,37]. In combination with molecular results and literature evidence, we believe that the Kashmir sample should represent the latest colonization rather than the original distribution. Additionally, according to haplotype and genetic difference analysis, we speculated that the introduced sample could have come from Zhejiang and surrounding coastal areas in China. This region was an important coastal city, with developed ports and frequent maritime trade, which could provide a good opportunity for its introduction [60,61]. In addition, Hap1 and its derived haplotypes have recently experienced rapid diffusion and have high environmental adaptability, which provides a good genetic basis for the colonization of this species after introduction [62]. Additionally, recent studies have suggested that genetic changes could underlie the greater invasiveness of bridgehead populations [63,64]. The Kashmir sample has already experienced unique genetic adaptive variation with a unique genotype (no haplotype shared with native samples), which indicated that they might be going through the process of establishing the bridgehead effect. However, this still needs further support from more samples and more molecular data. In combination with the results of ENM under the current conditions, we found that the suitable region of this species in Asia is mainly located in the second and third ladder regions of China, the Korean Peninsula, and Japan, and there is a banded highly suitable region on the southern edge of the Himalayas (Figure 5). The sample came from the Kashmir region, which is located on the southern edge of the Himalayas. Therefore, a suitable environment might provide good conditions for the survival of *C. fallax* in Kashmir.

In summary, we believe that *C. fallax* might have recently colonized Kashmir through trade or other accidental events. The introduction of highly adaptable genotypes and a locally friendly environment might together help with successful colonization in Kashmir. At the same time, the unique adaptive variation might help further establish the bridgehead effect and accelerate the invasion process.

### 4.3. Early Warning of Possible Invasion around the World

*C. fallax* is oligotrophic to legumes and has been recorded as a serious soybean pest in East Asia [32,33,34,37,65]. Therefore, the study of population dynamics and prediction of suitable habitats for this species will provide us with a direction for precise prevention and control of its invasion.

Most invasive species largely conserve their climatic niche, which has prominent implications for predicting future invasion risks [30]. The ENM results showed that except for East Asia, there is a wide range of highly suitable areas in southeastern North America, which could provide good environmental conditions for the survival of *C. fallax*. Additionally, this region is one of the main soybean-producing areas in the world [66], which indicates that it would provide an adequate food source for the invasion of *C. fallax*. Therefore, *C. fallax* presents a dramatically high invasion risk in southeastern North America. We suggest that the region should pay more attention to *C. fallax* in recent trade exchanges, especially for populations from North China and coastal areas that have had successful colonization experiences.

In addition, in the case of future global climate change, the ENM results showed that the predicted suitable habitat of the stalk-eyed seed bug will continue to move towards the poles, especially in the case of global warming, which will promote its movement process. The highly suitable area in Asia will increase significantly in the future, but the overlapping area with the soybean planted area showed a predicted decline. Therefore, we speculated that with the polar movement of the high-fitness area, the threat to soybean production in Asia by *C. fallax* will probably decrease in the future.

## 5. Conclusions

Here, we used population genetic methods and ecological niche modelling to reconstruct the invasion history of *C. fallax* and to predict its current and future potential distribution areas. We demonstrated that *C. fallax* has recently invaded Kashmir, and the invasive source has been identified as Zhejiang Province and the surrounding coastal areas in China. At the same time, *C. fallax* also showed a high risk of invasion in southeastern North America and might seriously threaten local soybean production. Our analysis suggested that the quarantine of *C. fallax* should be strengthened in highly suitable areas beyond its natural range to avoid it becoming the next globally invasive pest. Our study will help us deepen our understanding of the species and provide new insights into the monitoring and management of this agricultural pest.

## Figures and Tables

**Figure 1 insects-14-00433-f001:**
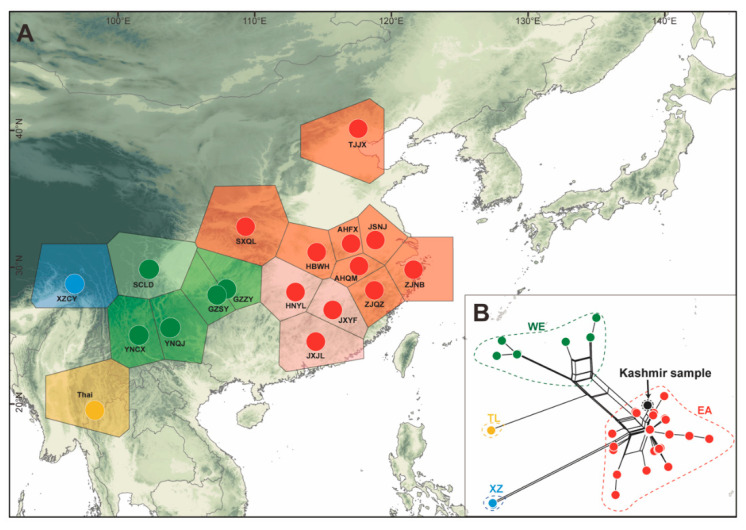
(**A**) Tessellation illustration of Bayesian analysis of population structure based on *COI*. Different colored dots represent different phylogenetic groups (red-EA; green-WE; yellow-TL; Blue-XZ). The detailed descriptions of site location abbreviations have been shown in Table 1. (**B**) Phylogenetic topology based on *COI* by the neighbour-net method. The dotted lines and arrow indicate *C. fallax* from Kashmir.

**Figure 2 insects-14-00433-f002:**
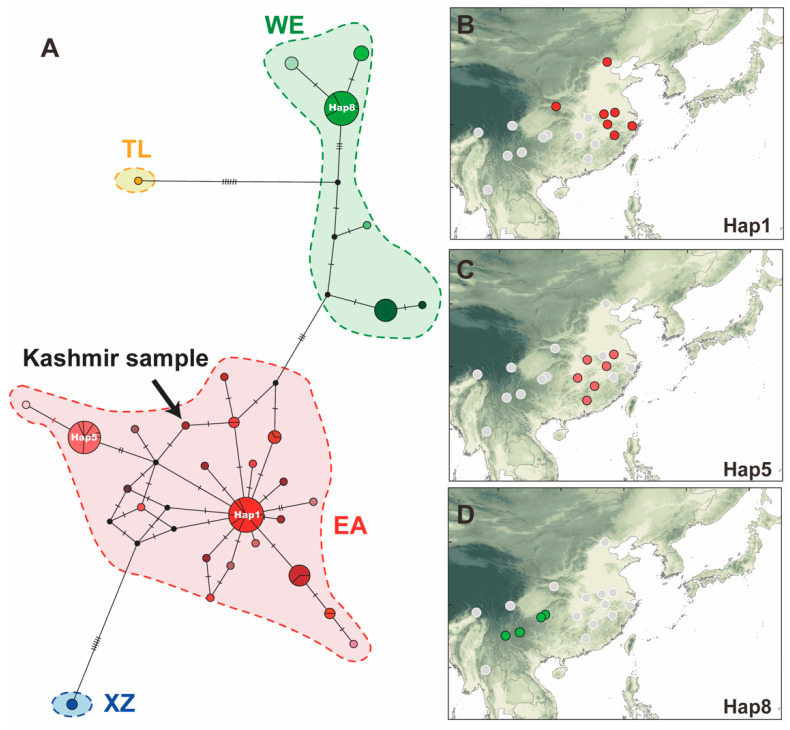
(**A**) Haplotype network of *Chauliops fallax*. Different dotted lines correspond to four phylogenetic groups (red-EA; green-WE; yellow-TL; blue-XZ). The arrow indicates *C. fallax* from Kashmir. (**B**) Geographical distribution of Haplotype 1. (**C**) Geographical distribution of Haplotype 5. (**D**) Geographical distribution of Haplotype 8.

**Figure 3 insects-14-00433-f003:**
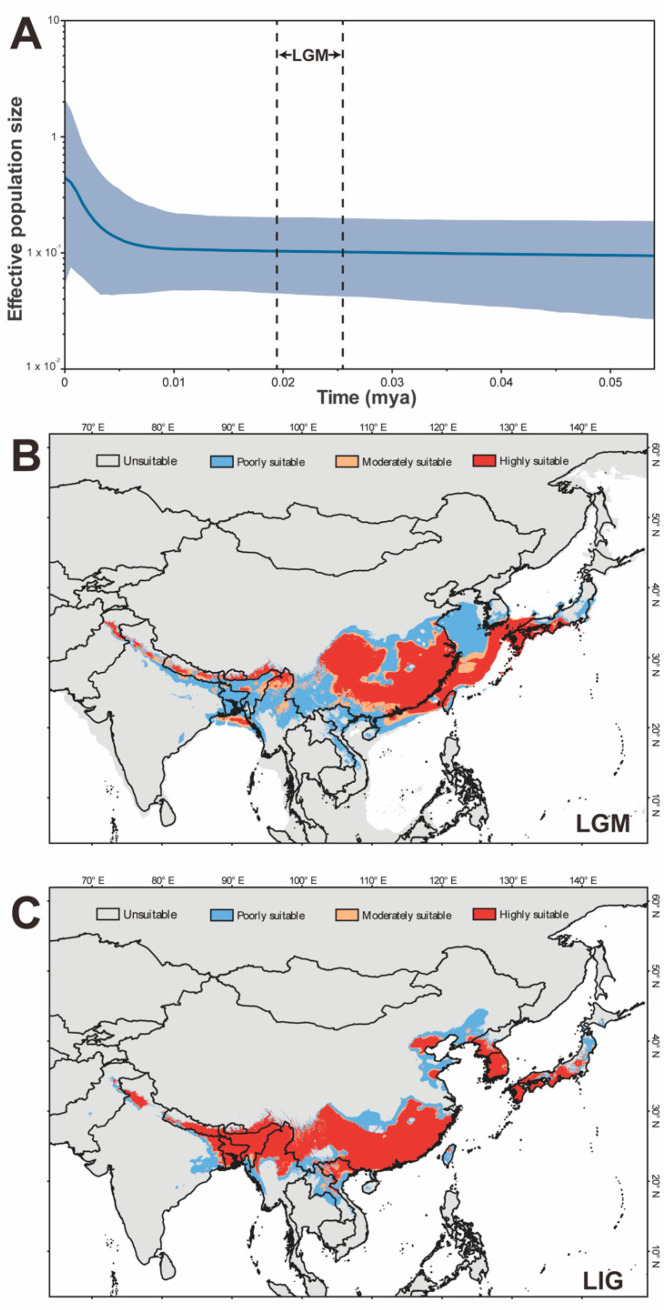
(**A**) Historical demographic changes in *Chauliops fallax*. Dark grey and lines represent 95% pseudo-confidence intervals. The X-axis represents time (millions of years ago), and the Y-axis represents the estimated scaled effective population size. The solid blue lines indicate the median value of the effective population size. The upper and lower limits of the light blue trend line represent 95% confidence intervals. (**B**) Modelled suitable areas of *C. fallax* in the LGM period. Colors indicate the probabilities of habitat suitability: unsuitable (0–0.177681), poorly suitable (0.177681–0.34739), moderately suitable (0.34739–0.404495), and highly suitable (0.404495–1), as below. (**C**) Modelled suitable areas of *C. fallax* in the LIG period.

**Figure 4 insects-14-00433-f004:**
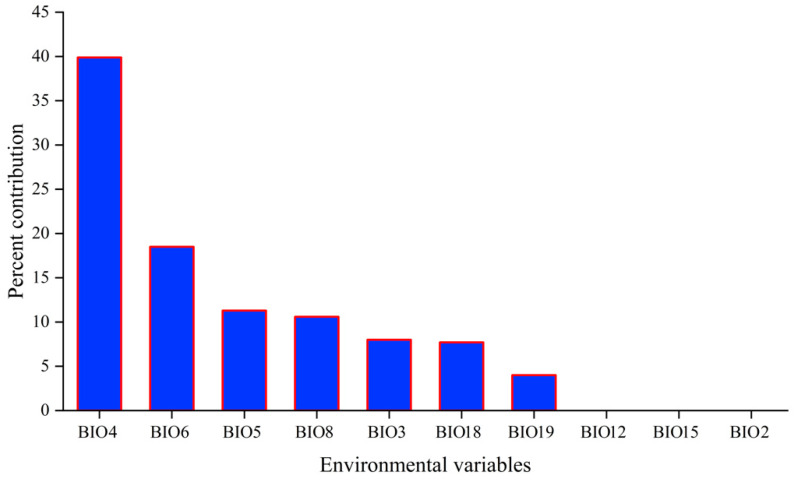
Relative contributions of different bioclimatic variables to the MaxEnt model for *Chauliops fallax*. (BIO2: mean diurnal range, BIO3: isothermality, BIO4: temperature seasonality, BIO5: max temperature of warmest month, BIO6: minimum temperature of coldest month, BIO8: mean temperature of wettest quarter, BIO12: annual precipitation, BIO15: precipitation seasonality, BIO18: precipitation of warmest quarter, BIO19: precipitation of coldest quarter).

**Figure 5 insects-14-00433-f005:**
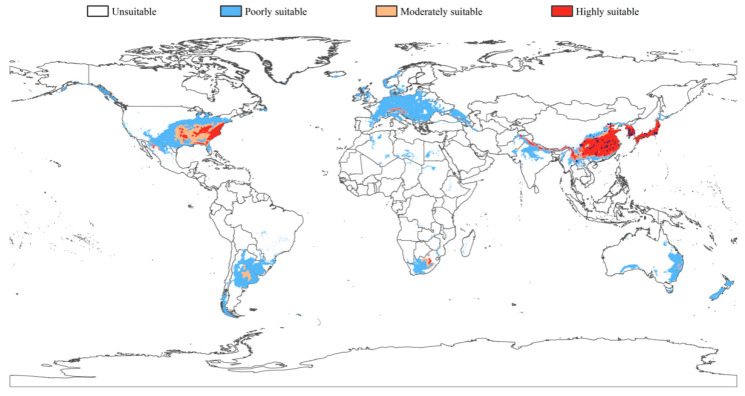
Global potential distribution of *Chauliops fallax* predicted by the MaxEnt model under present climate conditions with 83 occurrence records (blue points).

**Figure 6 insects-14-00433-f006:**
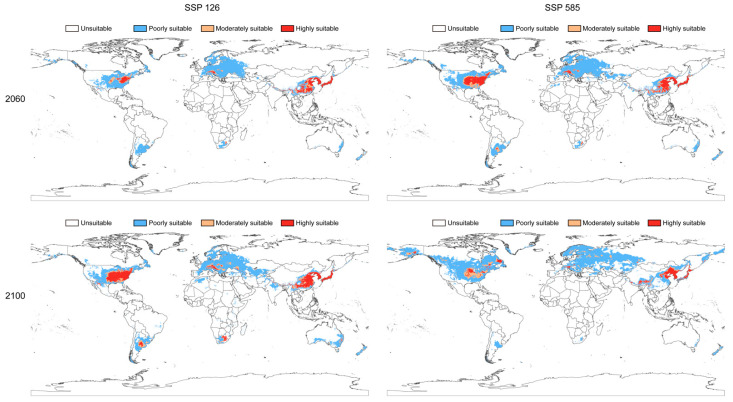
Global potential distribution of *Chauliops fallax* predicted by the MaxEnt model under future CanESM5 climate conditions. All six continents (Africa, Asia, Europe, North America, Oceania, and South America) would have potentially suitable habitat areas in the future. The overlapping area between the *C. fallax* greatest optimum suitable habitat area overlapped with the soybean planted area and decreased in the future for both SSP126 and SSP585 compared to the present climate conditions (Appendix A).

**Table 1 insects-14-00433-t001:** Nucleotide polymorphisms in each geographical population of *Chauliops fallax* for *COI*.

Population	Group	N	Location	Latitude/Longitude (°)	Hn	Hd	π
AHFX	EA	9	Zipeng Mountain National Forest Park, Feixi County, Anhui Province, China	31.7143/117.0013	3	0.417	0.00105
AHQM	EA	10	Chiling Village, Qimen County, Anhui Province, China	30.0485/117.5729	6	0.778	0.00274
HBWH	EA	7	Changxuan Mountain, Mulan Mountain, Wuhan City, Hubei Province, China	31.0975/114.4783	4	0.81	0.00421
HNYL	EA	2	Yuelu Mountain, Hunan Province, China	28.1855/112.9354	1	0	0
JSNJ	EA	10	Ginkgo Villa, Yuhuatai District, Nanjing City, Jiangsu Province, China	31.9542/118.7550	8	0.933	0.0039
JXJL	EA	9	Shrimp and bird pond, Jiulian Mountain, Jiangxi Province, China	24.5370/114.4180	2	0.222	0.00032
JXYF	EA	5	Shangdai Village, Shaxi Town, Yongfeng County, Jiangxi Province, China	26.8858/115.6316	1	0	0
SXQL	EA	4	Qianping Village, Xunyang City, Qinling Mountains, Shaanxi Province, China	32.9333/109.3017	3	0.833	0.00158
TJJX	EA	6	Chenglong Road, Jiulong Mountain, Jixian County, Tianjin, China	40.1312/117.5081	3	0.733	0.0019
ZJNB	EA	3	Yinzhou District, Ningbo City, Zhejiang Province, China	29.8172/121.5470	2	0.667	0.00105
ZJQZ	EA	6	Longjingkeng, Jiangshan City, Quzhou City, Zhejiang Province, China	28.3227/118.6739	3	0.6	0
Kashmir	EA	1	-	-	-	-	-
GZSY	WE	1	Suiyang County, Guizhou Province, China	27.9591/107.1823	-	-	-
GZZY	WE	10	Fengle Town, Wuchuan County, Zunyi City, Guizhou Province, China	28.3727/107.8602	4	0.81	0.00421
SCLD	WE	10	Wajiao Village, Luding County, Sichuan Province, China	29.8371/102.2098	2	0.2	0.00049
YNCX	WE	9	Duojiqiao Imperial Farm, Donghua Town, Chuxiong City, Yunnan Province, China	25.0107/101.4784	2	0.556	0.00088
YNQJ	WE	9	Wangsantun Primary School, Sanyan County, Qujing City, Yunnan Province, China	25.5378/103.7175	1	0	0
XZCY	XZ	2	Shangchayu Town, Chayu County, Tibet Province, China	28.7146/ 96.78446	1	0	0
Thai	TL	1	Soppong, Pangmapha District, Mae Hong Son Province, Thailand	19.5123/98.2555	-	-	-

Note: N, sample size; Hn, number of haplotypes; Hd, haplotype diversity; π, nucleotide diversity. Data Kashmir from NCBI (MN584895).

**Table 2 insects-14-00433-t002:** Predicted overlap coverage area (km^2^) between the soybean planted area and the potential distribution of *Chauliops fallax* under the present climatic conditions.

Suitability	Africa	Asia	Europe	North America	Oceania	South America	Total
Poor	2.35 × 10^5^	1.25 × 10^6^	1.20 × 10^6^	1.23 × 10^6^	1.85 × 10^5^	1.08 × 10^6^	5.18 × 10^6^
Moderate	6.32 × 10^4^	4.14 × 10^5^	2.32 × 10^4^	7.58 × 10^5^	7.71 × 10^3^	1.14 × 10^5^	1.38 × 10^6^
High	4.00 × 10^4^	2.04 × 10^6^	7.53 × 10^3^	4.87 × 10^5^	7.48 × 10^1^	5.72 × 10^2^	2.58 × 10^6^
Total	3.38 × 10^5^	3.71 × 10^6^	1.23 × 10^6^	2.48 × 10^6^	1.93 × 10^5^	1.20 × 10^6^	9.14 × 10^6^

## Data Availability

The dataset was deposited in Figshare (https://doi.org/10.6084/m9.figshare.22722388, accessed on 20 January 2023).

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
