# Peer review of "Out of East Asia: Early Warning of the Possible Invasion of the Important Bean Pest Stalk-Eyed Seed Bug Chauliops fallax (Heteroptera: Malcidae: Chauliopinae)"

_insects, 2023, doi:10.3390/insects14050433_

Round 1

Reviewer 1 Report

The article has a typical layout for research on invasive species, with particular use of the modeling method.

The use of genetic population research methodology is needed, but sampling from the entire natural range of the species is lacking.

There is a lot of speculation in the article that is poorly supported for various reasons by scientific facts.

The discussion in the first part (Lines 326-352) is actually a description of the results and the mentioned fragment should be moved there.

Conclusions are a repetition of the abstract/abbreviated description of the results and are not a summary of the article.

Author Response

Thank you very much for making a scrutiny into our manuscript.  Please refer to the attachment for the reply.

Reviewer 2 Report

Thank you for the opportunity to review this paper. I have only made minor comments to aspects of the paper. Overall I believe the science to be sound and worthy of publication; though after major editing to the text and some additional statistical analyses are undertaken. The use of mtDNA markers is somewhat dated in population genetics and would have been better supported by the use of a nuclear marker, SNPs in particular. What was the justification of using COI? There was no clear reason stated for its use over a nuclear marker, which is more appropriate for population genetic studies. mtDNA still has its place in phylogenetic studies, your study clear states it's a population genetic study. So this needs to be made clear as to why mtDNA was chosen over a nuclear marker. Also more statistical analyses could have been performed using the mtDNA data; a clear justification for the absence of some needs to be made or likewise a clear justification of why the current analyses and methods were used. Nevertheless, since population genetics only formed one part of the scientific methods undertaken, the use of mtDNA supported by the modelling was acceptable overall.  The climatic modelling was done well and bouyed the science undertaken. It is important that more care is given to providing a full description of labels used in figure and table captions as the abbreviations are too confusing otherwise. The manuscript uses many synonyms of mainstream English words, incorrectly. This is a problem with overall understanding of the manuscript because of poor language expression and incorrect use of words, and several rounds of English language editing will need to be undertaken to address this.  

Author Response

# To Reviewer2:

Q1: Thank you for the opportunity to review this paper. I have only made minor comments to aspects of the paper. Overall I believe the science to be sound and worthy of publication; though after major editing to the text and some additional statistical analyses are undertaken. The use of mtDNA markers is somewhat dated in population genetics and would have been better supported by the use of a nuclear marker, SNPs in particular. What was the justification of using COI? There was no clear reason stated for its use over a nuclear marker, which is more appropriate for population genetic studies. mtDNA still has its place in phylogenetic studies, your study clear states it's a population genetic study. So this needs to be made clear as to why mtDNA was chosen over a nuclear marker. Also more statistical analyses could have been performed using the mtDNA data; a clear justification for the absence of some needs to be made or likewise a clear justification of why the current analyses and methods were used. Nevertheless, since population genetics only formed one part of the scientific methods undertaken, the use of mtDNA supported by the modelling was acceptable overall.  The climatic modelling was done well and bouyed the science undertaken. It is important that more care is given to providing a full description of labels used in figure and table captions as the abbreviations are too confusing otherwise. The manuscript uses many synonyms of mainstream English words, incorrectly. This is a problem with overall understanding of the manuscript because of poor language expression and incorrect use of words, and several rounds of English language editing will need to be undertaken to address this.  

A1: Thank you very much for your recognition of our work. We have carefully revised the manuscript according to your valuable comments. Previous records have shown that the Chauliops fallax is native to East Asia and has not invaded other regions. However, a COI molecular fragment of C. fallax has been uploaded from Kashmir on NCBI (GenBank: MN584895) in 2020. Therefore, we hoped to explore whether the stalk-eyed seed bug was originally found in this area or whether it was the result of a recent invasion. In order to match the molecular data of Kashmir's sample, we used COI data as molecular markers here. According to your suggestions, we have added some statistical analyses of COI data in the manuscript and the description of the abbreviations has been provided in the legend. In addition, we have carefully revised the writing of the article. We sincerely hope the modification can conform to your request.

Q2: Line 13: “crucial”: inappropriate synonym of important.

A2: We have revised “crucial” to “important” in the whole manuscript.

Q3: Line 18-19: “The Kashmir sample was found to come from the recent invasion of populations in the coastal areas of southern China”: need to edit this sentence and add commas in appropriate location

A3: We have rewritten this sentence in the manuscript.

Q4: Figure 1: need to provide detailed descriptions of site location abbreviations. Link this with table 1 if needed

A4: Thanks for your valuable comment. We have linked the site location abbreviations to Table 1 and explained in the legend.

Q5: Line 136: “The Kashmir sample was a recent invasion sample and did not experience a long period of adaptive evolution (see result),” how do you know this? speculative statement unless you provide evidence

A5: We have removed this sentence from the manuscript.

Q6: line 150 “2.1.3. Historical population dynamics”: there are numerous other analyses that you could have done and justifying why you haven't undertaken them is important eg. why not compute Tajima's D?

A6: Thanks for your valuable comment. We used Bayesian Skyline Plot to reconstruct the effective population size fluctuations here, as this method could visually display the historical population dynamics. According to your comment, we have added the neutrality tests, i.e., Tajima’s D and Fu and Li’s D in the manuscript.

Q7: line 191 “2.2.2. Ecological niche modelling”: more information needs to be provided on the parameters modelled. What were suitable and unsuitable parameters modelled and why?

A7: We have added the sentence that: ‘Before modeling, the model-tuning procedure was performed using the R packages ‘ENMeval’. The model with the lowest delta Akaike information criterion (delta AIC) value was selected as the best model for the modeling.’.

The model with the lower delta AIC value means lower model complexity. Due to Wallacean shortfall, we could not collect all occurrence records. Therefore, it is not suitable for models with high complexity. Thus, the model with the lowest delta AIC value was suitable parameters.

Reference: Kass, J.M.; Muscarella, R.; Galante, P.J.; Bohl, C.L.; Pinilla-Buitrago, G.E.; Boria, R.A.; Soley-Guardia, M.; Anderson, R.P. ENMeval 2.0: Redesigned for customizable and reproducible modelling of species' niches and distributions. Methods Ecol. Evol. 2021, 12, 1602-1608, doi:10.1111/2041-210x.13628.

Q8: Figure 4: Need to write out the full descriptor of each env variable

A8: The full descriptor of each env variable has been added to the legend.

Round 2

Reviewer 1 Report

Well done, congratulations.